# Analysis of the Utilization and Prospects of CRISPR-Cas Technology in the Annotation of Gene Function and Creation New Germplasm in Maize Based on Patent Data

**DOI:** 10.3390/cells11213471

**Published:** 2022-11-02

**Authors:** Youhua Wang, Qiaoling Tang, Yuli Kang, Xujing Wang, Haiwen Zhang, Xinhai Li

**Affiliations:** 1Biotechnology Research Institute, Chinese Academy of Agricultural Sciences, Beijing 100081, China; 2Institute of Crop Sciences, National Key Facility for Crop Gene Resources and Genetic Improvement, Chinese Academy of Agricultural Sciences, Beijing 100081, China

**Keywords:** maize, CRISPR-Cas technology, germplasm, patent data

## Abstract

Maize (*Zea mays* L.) is a food crop with the largest planting area and the highest yield in the world, and it plays a vital role in ensuring global food security. Conventional breeding methods are costly, time-consuming, and ineffective in maize breeding. In recent years, CRISPR-Cas editing technology has been used to quickly generate new varieties with high yield and improved grain quality and stress resistance by precisely modifying key genes involved in specific traits, thus becoming a new engine for promoting crop breeding and the competitiveness of seed industries. Using CRISPR-Cas, a range of new maize materials with high yield, improved grain quality, ideal plant type and flowering period, male sterility, and stress resistance have been created. Moreover, many patents have been filed worldwide, reflecting the huge practical application prospects and commercial value. Based on the existing patent data, we analyzed the development process, current status, and prospects of CRISPR-Cas technology in dissecting gene function and creating new germplasm in maize, providing information for future basic research and commercial production.

## 1. Introduction

Improvement of the yield, quality, and stress resistance of crops is the goal of scientists and breeders. Conventional breeding, including cross-breeding and mutation breeding, has significantly improved crop yield and crop performance under various stresses over the last century, but traditional breeding is limited by bottleneck factors such as a lack of available germplasm resources, unfocused breeding objectives, difficult pyramiding of elite traits, a long breeding process, and high cost. Although transgenic technology is undoubtedly an alternative to improve crops with desirable traits, its major limitation is the strict safety regulatory procedures and the low public acceptance of genetically modified crops [1].

Genetic variation and diverse germplasm are the basis of crop breeding. Gene-editing technology can overcome the bottleneck of lack of germplasm resources in traditional breeding by introducing precise genetic alterations to the targeted genomes, efficiently accelerating the pyramiding of multiple traits [2]. Clustered Regularly Interspaced Short Palindromic Repeats (CRISPR)/CRISPR-associated proteins (CRISPR-Cas) technology has become a powerful tool for crop genetic improvement by efficiently and accurately modifying target genes, thereby accelerating plant breeding. In particular, different precise gene-editing tools such as base editing, prime editing, and gene targeting provide tools for the precise aggregation of multiple traits in an elite variety. CRISPR-Cas technology has been used widely in the analysis of key genes and the creation of new germplasm in rice, maize, soybean, and wheat, thus starting a new era of molecular design breeding in crops [3,4,5]. In the United States, gene-edited crops may take only 5 years from initial research to commercial production. Gene-edited products such as waxy corn, high oleic acid soybean, and high-GABA tomato are being produced commercially in the US, Japan, and other countries [6]. In 2022, the Guidelines for the Safety Evaluation of Gene Editing Plants for Agricultural Use (Trial) was issued by the Ministry of Agriculture and Rural Affairs of the People’s Republic of China, providing laws and regulations for the process of creating, safety evaluating, and commercially producing gene-edited crops.

Maize (*Zea mays* L.) is a food crop with the largest planting area and the highest yield in the world, and it plays a vital role in ensuring food security and alleviating energy crises. Due to limited elite germplasm resources and cross-incompatibility, conventional breeding methods are relatively insufficient and time-consuming, taking years to obtain ideal materials, and biotechnologies have proven to be attractive in the creation of excellent maize germplasm [7,8]. In particular, CRISPR-Cas can precisely and directly generate new lines with multi-trait aggregation within 2 years by modifying genes controlling different target traits [7]. Patenting is one of the most important indicators of technological innovation and new material creation, and it is an important foothold in market competition. In recent years, the research and applications of CRISPR-Cas gene-edited maize have become increasingly competitive, and increasingly related patents have been filed worldwide. The patent layout reflects the commercial prospects of gene-edited maize germplasm, indicating the importance of intellectual property protection in seizing technological innovation and market competitiveness in the future. This paper analyzes the research process, current status, and development trends in global gene-edited maize from the perspective of the relevant patent protections, which provides information for the macroplanning of future research and the commercialization of gene-edited maize materials.

## 2. Rapid Advances in Using CRISPR-Cas Editing in Maize

We first analyzed the relevant patent applications for CRISPR-Cas technology in maize (as of March 2022) using the global patent database search (https://analytics.zhihuiya.com/, accessed on 29 May 2022). We found 184 patents (Column A of Appendix A) on the use of CRISPR-Cas technology in analyzing gene function and the generation of new germplasm in maize worldwide. Considering the layout of the same patent content in different countries and regions as one patent, 123 different patents were obtained (Column B of Appendix A). Based on the number of patents per year, we identified an exploration period transitioning to a rapid development period. From 2015 to 2018, there were fewer than 10 patents per year (exploration period) (Figure 1). Since 2019, the related research has entered a rapid development period with more than 20 patents filed every year. Specifically, 48 patents were filed in 2021, accounting for 40.33% of the total patents (Figure 1). Of the 123 patents, 104 were filed by Chinese scientific research institutes, accounting for 84.55% of the total. In particular, great breakthroughs have been made in developing new maize germplasm and innovative cross-breeding methods using CRISPR-Cas technology (Appendix A). For example, the ZmRAVL1 maize gene and functional site, and the use thereof, was filed by China Agricultural University, providing a method of creating new maize germplasm with compact plant types suitable for dense planting and high yield, accomplished by editing the ZmRAVL1 gene [9]. Another example is the Artificial Creation of Maize Male Sterile Lines and Efficient Transformation Methods, created by the Institute of Crop Sciences, Chinese Academy of Agricultural Sciences, which present a new way of creating male sterile lines and the utilization of heterosis in maize [10]. These achievements indicate that the research and applications of gene-edited maize in China are at the forefront of the world. Moreover, waxy corn hybrids with higher yield were generated with CRISPR–Cas9 by editing waxy1 in 12 elite inbred maize lines. Importantly, these CRISPR–Cas9 waxy corn hybrids were not subjected to the regulatory oversight regarding genetically modified organisms in the United States, Argentina, Brazil, and Chile. A limited precommercial launch was conducted in the Midwestern United States in 2019 [11]. These major advances in CRISPR-Cas technology in maize have opened up a new era for the generation of new germplasm and elite varieties.

## 3. Effective and Precise Improvement of Agronomic Traits

Since genetic variation is the basis for the improvement of crop traits, crop breeding is aimed at developing elite varieties with high yield, good quality, and enhanced resistance to various stresses. CRISPR-Cas technology can accelerate the rapid pyramiding of multiple desirable traits by accurately modifying the specific target genes, thus forming the basis of a new era in efficient and precise crop breeding [12]. The 123 patents we found deal with genetic improvements in grain yield, grain quality, nutritional content, plant architecture, flowering time, fertility, haploid induction, and environmental stress resistance (Figure 2 and Column G of Appendix A). The total number of traits analyzed and edited in the 123 patents was more than 123 because several patents dealt with two or more traits.

### 3.1. Yield

High-yield maize has always been the first goal for breeders. However, the traditional breeding methods have limited effectiveness in improving maize yield, whereas the CRISPR-Cas technique has offered unprecedented advantages in enhancing crop yield with superior precision and speed [13]. There have been 25 patents related to the improvement of maize yield through the CRISPR-Cas editing of different target genes, accounting for 20.33% of 123 patents. Maize yield is determined by multiple genetic factors, such as effective ear number, kernel number per ear, and 100-kernel weight, as well as by environmental conditions [14,15]. The key genes controlling yield-related traits are, for example, ZmRLK7, ZmEREB102, ZmCEP1, UB2, UB3, ZmCO2, and GT1, and their editing can increase maize yield [16,17,18] (Appendix A). Another important way to improve maize yield in densely planted maize is closely related to plant height and the number and angle of leaves. For example, erect leaves are conducive to increased population density by maintaining photosynthesis and grain filling in dense planting [13]. Thus, modifying genes that control leaf angle, such as LG1 and ZmRAVL1, could generate new materials with compact plant sizes and increased population yield (Appendix A) [9,19]. Therefore, CRISPR-Cas technology provides new ways to develop high-yield germplasm resources.

### 3.2. Quality and Specialty Maize

Malnutrition caused by a lack of essential nutrients in cereal-based diets is a serious threat to millions of people worldwide. Therefore, a new challenge in maize breeding is improving the nutritional quality of grains, including specific proteins, carbohydrates, fatty acids, essential amino acids, and vitamins [20]. In recent years, progress has been made in mining quality-related genes and creating new specialty maize germplasm using CRISPR-Cas technology. For example, we found 25 patents (accounting for 20.33% of the total) on the improvement of maize quality by editing different genes that encode metabolic enzymes (including granule-bound starch synthase, starch synthase, starch branching enzyme, glutamic imine methyltransferase, and betaine aldehyde dehydrogenase), functional proteins (such as PPR protein and gliadin 20S proteasome subunit), and transcription factors; these patents include the creation of the new germplasm with improved grain waxiness, aroma, amino acid content, folate content, and starch components and content (Appendix A). For example, CRISPR-Cas editing Waxy1, SHRUNKEN2, and ZmBADH2a/b can effectively generate waxy, sweet, and aromatic maize grain, respectively [11,21,22], which may meet increasing demands for special tastes and flavors from the consumer market.

Deficiencies in the essential vitamins and nutrients (hidden hunger) needed for optimal health are the main cause of many diseases (such as diabetes, cardiovascular ailments, and cancer), as well as obesity. Using biotechnology in breeding provides effective strategies to alleviate hidden hunger by increasing the content of micronutrients in crops [23]. For example, the knockout of ZmGFT1 can significantly increase the content of 5-methyltetrahydrofolate in maize, which provides a new way to alleviate folic acid deficiency via biofortification (Appendix A). These findings and patents provide an important theoretical basis, as well as technical strategies, for the improvement of maize quality.

### 3.3. Male Sterility

Heterosis is an effective way to increase crop yield, and it has been used most successfully in breeding maize hybrids. The breeding of male-sterile lines is an important prerequisite for heterosis utilization in crops, which can obviate the need for manual emasculation, reduce labor costs, and improve seed purity and yield, and thus, it has very important breeding and commercial value [24,25] In maize production at present, most male-sterile lines have some limitations due to susceptibility to diseases, unstable fertility, and extremely complicated creation process. Thus, the creation and application of new male-sterile lines is an important goal in maize production, and the identification and functional analysis of fertility-related genes is the key prerequisite for creating new male-sterile lines [26,27]. Recently, using CRISPR-Cas technology, a number of male-sterility genes, encoding transcription factors, key enzymes, and other proteins, were annotated and utilized to generate a series of new male-sterile materials in maize [25]. There are 13 patents related to the identification of fertility-related genes and the creation of new male-sterile lines in maize using CRISPR-Cas technology (Appendix A). In particular, based on gene editing technology, a simple method for the one-step creation of male-sterile lines associated with the matching maintainer line (an artificially created maize male-sterile line and an efficient transfer method) was established; this method overcomes the bottlenecks of traditional breeding technologies and will promote the industrial application of third-generation crossbreeding techniques [10]. Therefore, CRISPR-Cas technology underpins new opportunities for mining male sterility genes, creating male-sterile lines, and enhancing heterosis utilization in maize.

### 3.4. Stress Resistance and Herbicide Resistance

Environmental stresses such as drought, high temperature, low temperature, flooding, pests, diseases, and soil salinity seriously affect the yield and quality of maize. Therefore, the genetic improvement of maize yield is closely related to the enhancement of stress resistance. Recent studies have confirmed that plant stress resistance is a quantitative trait controlled by the interactions of multiple genes and environmental factors, and biotechnology breeding is an inevitable choice to improve maize resistance to biotic and abiotic stresses [28]. In recent years, many key genes involved in stress responses have been identified and used to generate new maize materials with enhanced stress tolerance with CRISPR-Cas technology. There are seven patents related to the generation of new germplasm with improved resistance to northern leaf blight, straw disease, rough dwarf disease, ear rot, stem rot, and sheath blight by editing Ht1, NLB18, ZmSIZ1a, ZmSIZ1b, ZmFhb1, and m00001d010255, respectively [29] (Appendix A). For abiotic stress, there are nine patents related to the improvement of maize tolerance to abiotic stresses such as drought, low temperature, and high temperature (Appendix A) by activating the expression of ZmBG1 [30] or the knockouts of ZmEREB102, DRK, PP84, CPK2, AL14, ZmbZIP68, ZmCIPK15, and ZmAAPa [31,32,33]. These studies show that the development of new maize materials with enhanced stress resistance via CRISPR-Cas technology has strong potential applications.

With the constantly increasing level of intensity and mechanization in maize production, the application of herbicides is a primary and effective weed control strategy due to their superior efficacy, reduced labor demand, relatively low cost, and increased crop yield. In recent decades, herbicide-resistant, genetically modified crops brought a revolution in weed management systems, with such crops becoming the most significant transgenics [34,35,36]. In particular, there have been many reports of using CRISPR-Cas technology to create desired herbicide-resistant germplasm by editing genes related to herbicide resistance such as acetolactate synthase (ALS), acetyl-CoA carboxylase (ACCase), and 5-enolpyruvate oxalate-3-phosphate synthase (EPSPS) in different crops, which can facilitate the flexible use of robust, non-selective, and broad-spectrum herbicides [37,38]. In maize, two patents have reported on the new herbicide-resistant germplasm materials created by CRISPR-Cas editing ALS1/2 and EPSPS (Appendix A), which have good commercial prospects in maize production. Therefore, it will be of great importance to apply this technique to identify more genes related to herbicide resistance and create new herbicide-resistant germplasm in maize.

### 3.5. Plant Architecture

Maize plant architecture traits, including plant height, ear position, leaf angle, and internode length, determine a plant’s yield by affecting the canopy structure, photosynthetic efficiency, and planting density, as well as resistance to lodging and various stresses. Thus, one of the most effective methods to increase maize yield per unit area is by breeding elite hybrids with optimized plant architecture [39]. CRISPR-Cas technology has shown its unique advantage in creating ideal plant-type materials in maize. There are at least 25 patents related to editing the key genes that control maize plant height and leaf angle (Appendix A). First, plant height is closely associated with the number of nodes and the length of each internode, and the homeostasis and signal transduction of phytohormones such as gibberellins and brassinosteroids play important roles in the regulation of plant height [40,41]. In maize, editing the key genes controlling the homeostasis of gibberellin (ZmGA20ox3, ZmGA20ox5, ZmGA2ox3, and ZmGA3ox1), Zm-BR1, and ZmPIF3s can quickly generate multiple dwarf plant types [42] (Appendix A). Leaf angle is an important agronomic trait in the breeding of high-yield varieties by affecting light interception, photosynthetic efficiency, and planting density [43,44]. In maize, ZmRAVL1 regulates endogenous brassinosteroid content by activating brd1 expression, and the knockout of ZmRAVL1 can create upright plant architecture with reduced leaf angle and increased yield under high planting density [9] (Appendix A). These studies indicate that CRISPR-Cas techniques can rapidly optimize plant architecture by precisely editing the key genes, which provides an effective strategy for breeding densely planted and high-yield varieties.

### 3.6. Flowering Time

Flowering is an important trait for optimum crop life cycles and yields; it marks the transition from vegetative to reproductive growth and influences adaption to environmental stresses. Hence, controlling appropriate the flowering time is a major goal of breeders in developing elite varieties with better adaptations to local environmental conditions and climatic changes [45,46]. Flowering time is a quantitative trait controlled by multiple genes, and an increasing number of key genes from nuclear factor-Y and the CCT transcription factor family have been identified in maize [47,48]; they are the ideal target genes for CRISPR-Cas technology to create maize germplasm suitable for different ecological regions. At least 17 patents (accounting for 13.82% of the total) have reported new maize materials with improved flowering times, created by editing key genes that control the maize flowering time, including ZmFKF1, ZmGA20ox3, ZmDTX3.1, ZmPHYC1/2, and several CCT transcription factor genes (ZmCCT10, GRMZM2G044126, GRMZM5G878561, GRMZM2G331652, GRMZM2G068943, and GRMZM2G172297) [49] (Appendix A). These CRISPR-Cas manipulations provide a feasible strategy for precisely breeding new maize varieties suitable for different environments.

### 3.7. Haploid Induction

Conventional breeding made slow progress in breeding maize varieties due to a long breeding cycle, a lack of quality resources, and gene linkages/drag. Doubled haploid technology has become an integral part of modern maize breeding due to its speed, high efficiency, flexibility, and genetic benefits over conventional technologies, and it has huge commercial value and major breeding prospects [50,51,52]. However, the genetic basis of haploid induction remains unclear, and the selection of lines with high-frequency haploid induction lines is difficult. Therefore, cloning new genes involved in haploid induction is important for further elucidating its genetic mechanism and developing novel lines. Recently, CRISPR-Cas technology was used to identify genes related to haploid induction and create high-rate haploid inducers by editing the pollen-specific phospholipase genes [53,54,55]. Five patents reported that multiple haploid inducers were generated in maize by editing ZmPLD3, ZmDMP, ZmPLA1E, ZmPLA1, and GRMZM2G471240 [53,55] (Appendix A), suggesting that CRISPR-Cas technology will bring new opportunities and broad application prospects in the creation of maize lines with a high rate of haploid induction.

### 3.8. Other Traits

Gametophytic cross-incompatibility has been used to provide reproductive isolation in commercial maize production, and understanding its regulatory mechanism is important in revealing the process of fertilization and improving seed production and the genetic purity of maize products [56]. Recently, patent CN108794610B reported a new gene resource for studying maize incompatibility by CRISPR-Cas editing ZmGa1S [57], which can be utilized in the process of maize breeding and seed production (Appendix A).

Lodging seriously affects maize yield. Patent CN110627888B reported that the knockout of Stiff1 by CRISPR-Cas technology can significantly increase stalk strength, thus providing new germplasm and a method for improving maize lodging resistance [58] (Appendix A). Moreover, two patents (CN112680472A and CN113817033A) have asserted that a knockout of ZmELF3.1, ZmSBP20, ZmSBP25, and ZmSBP27 can create germplasm with improved root systems and aerial root assemblages (Appendix A), providing a theoretical basis and technical strategies for the improvement of maize root systems.

Taken together, CRISPR-Cas technology provides a powerful tool for the genetic improvement of multiple traits in maize.

## 4. Ideal Target Genes Are the Key to Generating New Germplasm

To improve crop traits using gene editing technology, it is necessary to understand the functions and molecular mechanisms of the key genes controlling specific target traits. Selecting ideal target genes is the key to precisely creating the desired germplasm materials. In this paper, the proteins encoded by corresponding target genes are classified according to their function descriptions in 123 patents (Figure 3, Column F of Appendix A). We found that 43 genes (accounting for 34.96% of the total) encode multiple members from different transcription factor families, such as CCT, SBP, bZIP, bHLH, and TAG, whereas 20 genes (accounting for 16.26% of the total) encode various regulatory proteins, including protein kinase, calcium-binding proteins, protein phosphatase, histone acetyltransferase, hormone peptide, receptor protein, PcG protein, DELLA protein, and phytochrome C. It has been reported that transcription factors and regulatory proteins participate in plant development processes and stress responses by regulating the expression of a large number of downstream genes [59,60], making them the ideal target genes for the creation of new germplasm through gene-editing technology. Thirty genes (accounting for 24.39% of the total) encode enzymes such as starch synthase, galactose oxidase, dihydroflavonol-4-reductase, lipoxygenase, gibberellin oxidase, acetyl lactate synthase, methionine synthase, and cytochrome oxidase, which are involved in the synthesis or metabolism of specific substances and phytohormones, cell signal transduction, redox homeostasis, grain quality, male sterility, flowering time, plant type, and stress resistance of maize. Six genes are involved in the post-translational modification of proteins and encoding, e.g., protein S-acyltransferase, F-box protein, and E3 ligase, which regulate the activity, stability, and function of specific proteins. In addition, five genes encode transporters (ATP-binding cassette transporter, malate transporter, ABCG-type transporter, and sulfate transporter), fourteen encode other functional proteins (guanosine diphosphate dissociation inhibitor, membrane protein, PPR protein, gliadin, plastid ribosomal assembly factor, and glycoprotein), and four are associated with unknown functional proteins; these transporters and other functional proteins play regulatory roles in multiple physiological processes and agronomic traits. Thus, the selection of a suitable target gene is most important for the precise creation of excellent germplasm resources using gene-editing technology.

## 5. Prospects of Gene-Edited Maize

### 5.1. Global Commercial Prospects of Gene-Edited Maize

CRISPR-Cas technology overcomes the constraints of traditional breeding methods and provides new opportunities for ensuring world food security by precisely creating new varieties with high yield, improved quality, and enhanced stress resistance. It has been estimated that the market for gene editing will exceed USD 5 billion in 2025. Although gene-edited products have been commercialized in different countries, their global marketing also depends on the regulatory policies of each country. Without scientific consensus and practical regulatory systems, gene-edited crops face similar situations to genetically modified (GM) crops, and their future commercial production and public acceptance will likely be subject to unpredictable restrictions [61]. Encouragingly, many countries, such as the United States, Canada, Australia, Japan, Argentina, and Brazil, tend to adopt product-oriented regulatory policies, treating gene-edited crops without foreign gene introduction as non-genetically modified organisms (non-GMOs), generally allowing them to enter the market without safety evaluation. Moreover, significant gene-editing policy changes have also been enacted in Europe (https://ihsmarkit.com/research-analysis/significant-gene-editing-policy-changes-in-europe.html, accessed on 10 May 2022). In the United Kingdom, Parliament approves statutory instrument making gene editing trials easier to conduct accessed on 14 March 2022. These regulatory policy changes have greatly promoted the commercialized production of gene-edited crops, with more than 150 gene-edited crops entering the market in the United States. Recently, CRISPR-Cas waxy corn hybrids with agronomy superior to introgressed hybrids were generated by editing a waxy allele in 12 elite inbred maize lines [11]. Then, the USDA concluded that these waxy corns were not to be regulated by the APHIS regulations concerning GMOs (https://www.aphis.usda.gov/biotechnology/downloads/reg_loi/17-076-01_air_response_signed.pdf, accessed on 12 January 2018), and a limited precommercial launch was conducted in the Midwestern United States in 2019. Subsequently, Argentina, Brazil, and Chile similarly confirmed that CRISPR–Cas9 waxy corn is outside the scope of regulatory oversight for GMOs in these countries [11]. In January 2022, the Guidelines for the Safety Evaluation of Gene Editing Plants for Agricultural Use (Trial) was issued by the Ministry of Agriculture and Rural Affairs of the People’s Republic of China, greatly simplifying the regulatory policy for gene-edited plants; these guidelines will promote the field testing, safety evaluation, variety certification, and commercial production of gene-edited crops in China in the future.

### 5.2. Opportunities and Challenges of Gene-Edited Maize in China

China has been a world leader in CRISPR-Cas-edited crops, such as rice, maize, wheat, soybean, and others. Among the 123 global patents related to the use of CRISPR-Cas technology in maize, there are 104 patents (accounting for 84.55%) filed from China, documenting a large number of novel, elite germplasm materials regarding sterile lines, haploid inducers, plant types, high yield, improved quality, and enhanced stress resistance (Appendix A). In particular, a series of breakthroughs have been made in the creation of third-generation male-sterile maize lines, as well as heterosis utilization and haploid induction. Therefore, China should seize the new opportunity of gene-editing technology to control global crop variety breeding and commercial production in the future.

Firstly, the theoretical breakthroughs and technological innovation of gene editing in crops should be acknowledged. CRISPR-Cas technology can efficiently and precisely create new germplasm by modifying multiple key genes that control the target traits, thus providing new opportunities for the smart molecular breeding of crops [12]. Although China has made fruitful achievements in the research and application of CRISPR-Cas in crop editing, it must further develop new gene-editing technology with independent intellectual property rights, identify important genes with breeding values, and clarify the regulatory mechanisms governing agronomic traits. This research will provide the theoretical basis, technical support, and key gene resources for the creation of new, elite maize varieties.

For China, it is imperative to seize the opportunity of gene-editing technology to promote the development and international competitiveness of the national grain industry. Firstly, further integration of gene-editing technology with genomic selection, smart breeding, and conventional breeding technologies will realize the leap from the era of molecular breeding to the new era of intelligently designed breeding, which will accelerate the process of breeding excellent maize germplasm.

Secondly, the application of male-sterile lines in maize breeding should be strengthened. The use of male-sterile lines can avoid manual or mechanical emasculation, reduce production costs, and improve the purity and yield of seed, which has broad industrial prospects and commercial value worldwide. With the combination of gene editing, pollen inactivation, fluorescence seed screening, herbicide resistance screening, and conventional breeding methods, excellent multi-control male-sterile lines will be developed in the future. In China, 13 patents related to the creation of new male-sterile maize lines have been filed, and it is urgent to promote their field trials and utilization in breeding.

Thirdly, gene-editing technology provides new opportunities for studying the mechanism of haploid induction and the creation of haploid induction lines. Conventional maize breeding requires continuous self-crossing for eight or more generations to produce an inbred line, whereas haploid technology can create homozygous inbred lines over two generations in one year, which has become one of the three core technologies of modern maize breeding. Recently, many original studies have been conducted in China with respect to mining key genes controlling haploid induction, establishing efficient systems for haploid induction and creating haploid inducers (S1). These studies provide the theoretical basis and technical support for promoting the transformation of maize breeding technology in China.

It is worth noting that many reported transgenic plants have been tested for their potential breeding value only in the laboratory or greenhouse conditions, and the lack of field testing has resulted in the poor availability of some products [8]. In maize, the utilization value of 1671 genes (approximately 4.4% of the genome) has been evaluated through field tests, but only 22 genes have been proven to possess breeding values [8]. Therefore, field testing is indispensable in evaluating whether gene-edited maize can become an important germplasm resource for breeding. Although many CRISPR-Cas-edited maize materials have been generated, due to regulatory policies and the lack of reliable field trials, only CRISPR-Cas-edited waxy maize hybrids have been approved for commercial production in the United States [11]. So far, there is no report on their field testing in China. Recently, the Guidelines for the Safety Evaluation of Gene Editing Plants for Agricultural Use (Trial) were issued to regulate the research and field experiments on gene-edited crops, including the regulatory framework, safety evaluations, variety approval, and commercial production. According to these guidelines, most CRISPR-Cas-edited maize materials that do not involve environmental safety and food security can apply for a safety certificate in 2–3 years; then, they can be certified by the Variety Certification Committee within 2–3 years. Thus, this system greatly accelerates the process of breeding new varieties and their commercial production.

## Figures and Tables

**Figure 1 cells-11-03471-f001:**
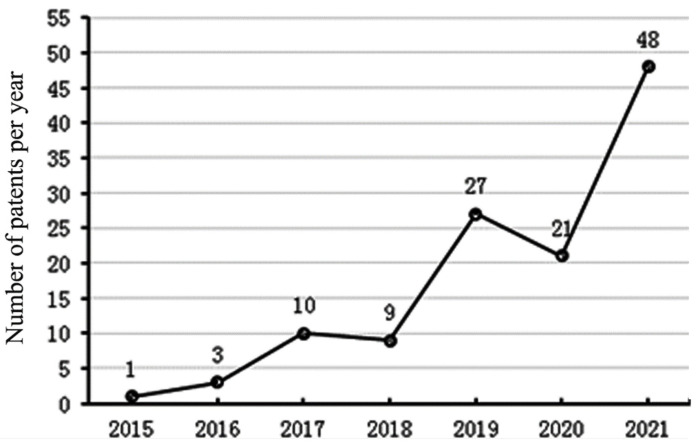
The number of patents per year arising from research on CRISPR-Cas-edited maize from 2015 to 2021.

**Figure 2 cells-11-03471-f002:**
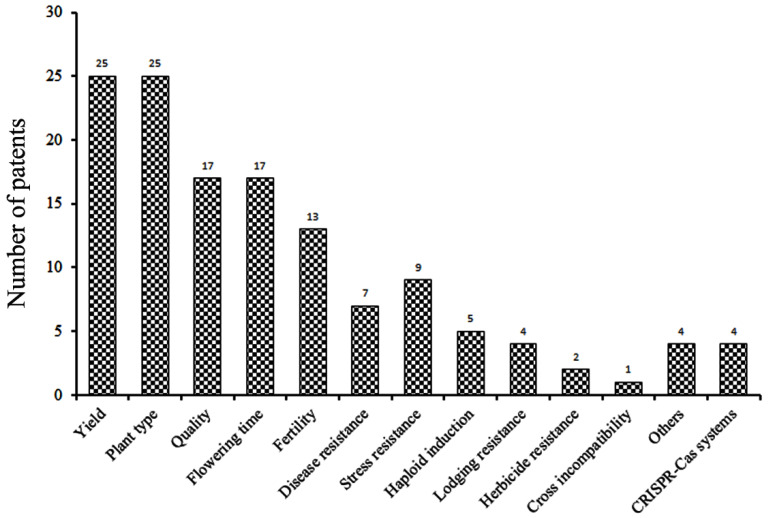
The number of patents related to the improvement of maize traits using CRISPR-Cas technology.

**Figure 3 cells-11-03471-f003:**
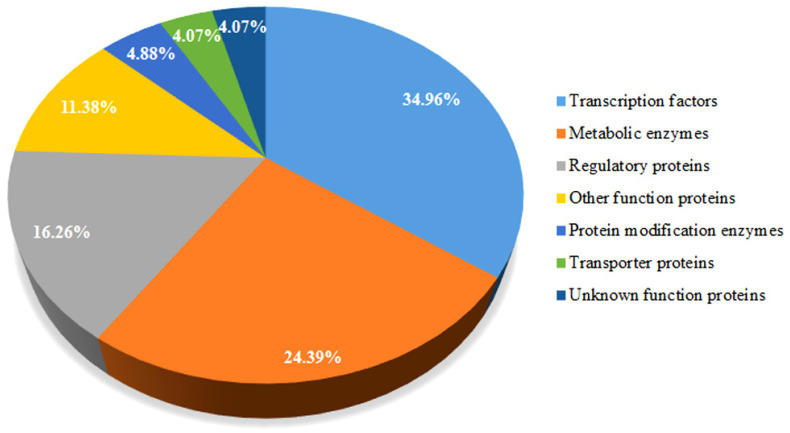
Classification of proteins encoded by the target genes.

## Data Availability

Not applicable.

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
