# Peer review of "Analysis of the Utilization and Prospects of CRISPR-Cas Technology in the Annotation of Gene Function and Creation New Germplasm in Maize Based on Patent Data"

_cells, 2022, doi:10.3390/cells11213471_

Round 1
Reviewer 1 Report
The author analyzed the development process, current status, and prospects of CRISPR-Cas technology in dissecting gene function and creating new germplasm in maize in China based on patent data. It's strongly suggested that the publication of these genes in related patent should be cited.
In the manuscript entitled “Analysis of the utilization and prospects of CRISPR-Cas technology in creating new maize germplasm based on the patent data”, authors analyzed the development process, current status, and prospects of CRISPR-Cas technology in dissecting gene function and creating new germplasm in maize in China based on patent data. The study provides useful information of CRISPR-Cas technology in maize breeding in China. However, the flowing points should be addressed before accepting for publishing in Cells.
Major concerns:
1. Re-consider the title of manuscript. These patents were focused on the use of CRISPR-Cas technology in analysis of gene function and generation of new germplasm in maize, not only on creation of germplasm.
2. It's strongly suggested that the publication of these genes in related patent should be cited.
Minor concerns:
Abstract
1. Line 12, delete [it]
2. Line 13-14, …. ineffective in breeding new maize varieties -> …. In maize breeding
3. Line 16, delete [technology]
4. Line 17, With this technology -> Using CRISPR-Cas
5. Line 20, this paper analyzes -> we analyze
Introduction
1. It’s strongly suggested to re-writing the introduction, it should be at least three paragraph to introduce the merits and demerits of traditional breeding methods and CRISPR-Cas and give examples that have been reported, briefly introduce the CRISPR-Cas.
2. Line 35- 38, citation or URL site should be listed.
3. Line 27, To improve -> Improvement of
4. Line 39, delete [it]
5. Line 45-49, check and re-write the sentence
6. differentiate the title of first and second column of supplementary table 1
7. Line 66, the publication of [Teosinte ligule allele narrows plant architecture and enhances high-density maize yields] should be cited. Likewise, it's strongly suggested that the publication of these genes in related patent should be cited in the main text.
8. It’s strongly suggested author re-organize the part of [4.2 Opportunities and challenges of gene-edited maize in China], summary and shorten the main points. Line 279-281, 289-291,315-319 should be introduced in introduction
Reviewer 2 Report
I thank the authors of “Analysis of the utilization and prospects of CRISPR-Cas technology in creating new maize germplasm based on the patent data” work. Few comments/suggestions are given below pointwise will help in improving the text and better presentation of the manuscript:
- I suggest mentioning brief information about CRISPR/Cas in the introduction section and adding a comparison between CRISPR and standard transgenic methods.
- It is better to provide a caption for figure 1.
- Line 80: “In this paper, we analyzed the protein function characteristics of 123 maize genes 80 related to 123 patents (Figure 2, Supplementary Table 1)”. Please describe what Bioinformatics methods you used for this analysis.
- The caption is required for figure 2 and figure 3.
- It is better to have supplementary table 1 in different colors for different agronomic traits to make it easier to read.
- Please also add the English translation of the patent titles written in Chinese.
- Row 162 in the table for patent number CN107298701B in the agronomic trait column, the sentence is incomplete: “Quality a….”.
- Please prepare a table for the proteins mentioned in the text, in which the exact function of the protein is written. When a review paper is written, it is expected to access comprehensive information about the topic easily. For example, ZmBGl is a membrane protein. What is its function that has increased drought resistance and yields? Please describe their function as you explained ZmRAVL1 in the “Plant architecture” section. In my opinion, for a review paper, it is better to add more information, thus, the researcher can easily find the desired gene by reading the paper.
Considering the above, I recommend this manuscript for publication after applying the changes.
Reviewer 3 Report
In manuscript, Wang et al firstly organized patents regarding maize engineering based on CRISPR-Cas technology. Then according to those data, they summarized the development of CRISPR-Cas application in maize and analyzed target genes related to new maize germplasm generation. Furthermore, improvements of agronomic traits of maize after engineering were introduced in detail and prospects of gene editing in maize was proposed. However, the demonstration of CRISPR-Cas technology in maize editing is kind of too general here. The following are some major and minor points that are suggested to be addressed.
1. For CRISPR-Cas technology, there are several types including type I/II/III/V et al. And the popular Cas proteins including Cas9/ Cas12a/Cas13 et al. They could be used for gene knockout/knockdown/knockin/activation et al. Therefore, when summarize CRISPR-Cas application, it’s better to classify clearly about which Cas protein used, what type of editing performed, et al.
2. In subtitle 2, the authors performed statistics based on enrichment analysis of all the target genes from patents regarding CRISPR-Cas based maize engineering. However, since the agronomic traits for different target gene varies, it could be too general to roughly perform the enrichment analysis based on the whole dataset. To classify gene sets based on agronomic traits first and then perform enrichment analysis is recommended.
3. There should be figure legends for each figure.
4. Line249, Canada is double used.
Round 2
Reviewer 2 Report
I thank the authors of “Analysis of the utilization and prospects of CRISPR-Cas technology in creating new maize germplasm based on the patent data” for the revised manuscript. A few changes are still needed:
- In my opinion, the title needs to be corrected. It is not appropriate to use the “analysis” twice.
- I did not find the captions for the figures in the revised manuscript.
- If desired, using colored rows based on the agronomic traits in supplementary table 1 is better than using colored fonts. The colored row helps the reader find the desired gene faster and makes it easier to read.
Author Response
We are grateful for your careful reviewing our manuscript and providing good suggestions. We have carefully resized the manuscript according to your comments. Now, we resubmit our revised manuscript and provide our responses to the referees’ comments point by point.
Kind regards,
Haiwen
Reviewer 2
I thank the authors of “Analysis of the utilization and prospects of CRISPR-Cas technology in creating new maize germplasm based on the patent data” for the revised manuscript. A few changes are still needed:
- In my opinion, the title needs to be corrected. It is not appropriate to use the “analysis” twice.
Thanks. We have revised it.
- I did not find the captions for the figures in the revised manuscript.
Thanks. We have added the captions for three figures in this revised manuscript
- If desired, using colored rows based on the agronomic traits in supplementary table 1 is better than using colored fonts. The colored row helps the reader find the desired gene faster and makes it easier to read.
Thanks, we have revised the Table 1 according to your suggestions.
Reviewer 3 Report
Based on the revised version of the manuscript, here are some following points I’d like to mention.
1. If the technology used is CRISPR-dCas9 mediated gene activation system, I suggest use CRISPR-dCas9 rather than CRISPR-Cas9.
2. According to the title which emphasizes analysis of the utilization of CRISPR-Cas technology in maize, I suggest adding a main figure or table to summarize different CRISPR-Cas system used in maize editing from patents.
3. The subtitle 1 is suggested to use “Rapid advances in using CRISPR-Cas editing in maize according to patent application”. In this paragraph, all the analysis are based on relevant patents published. If we use the original subtitle, I will suggest include the history of CRISPR-Cas editing in maize based on all the peer reviewed publications.
4. In Table S1, column “modification type of target gene”, some information are missing for example row 145/163 et al.
5. I am still not able to see the figure legends. But I guess the author exchanged figure 2 and figure 3. In that case the location of the two figures also needs to be changed.
Author Response
Dear editor and Reviewers
We are grateful for your careful reviewing our manuscript and providing good suggestions. We have carefully resized the manuscript according to your comments. Now, we resubmit our revised manuscript and provide our responses to the referees’ comments point by point.
Kind regards,
Haiwen
Reviewer 3
Comments and Suggestions for Authors
Based on the revised version of the manuscript, here are some following points I’d like to mention.
- If the technology used is CRISPR-dCas9 mediated gene activation system, I suggest use CRISPR-dCas9 rather than CRISPR-Cas9.
Thanks, we have resized it in table 1 according to your suggestion.
- According to the title which emphasizes analysis of the utilization of CRISPR-Cas technology in maize, I suggest adding a main figure or table to summarize different CRISPR-Cas system used in maize editing from patents.
Thanks. we have checked the CRISPR-Cas technology used in patents and added related data in Table1 (Column H), and almost reported patents were used CRISPR-Cas9 system.
- The subtitle 1 is suggested to use “Rapid advances in using CRISPR-Cas editing in maize according to patent application”. In this paragraph, all the analysis are based on relevant patents published. If we use the original subtitle, I will suggest include the history of CRISPR-Cas editing in maize based on all the peer reviewed publications.
Thanks. We have changed it according to your suggestion.
- In Table S1, column “modification type of target gene”, some information are missing for example row 145/163 et al.
Thank your reminders. We have checked these data and revised them.
- I am still not able to see the figure legends. But I guess the author exchanged figure 2 and figure 3. In that case the location of the two figures also needs to be changed.
Thanks. We have checked and added figure legends.